# Radiologist-Trained and -Tested (R2.2.4) Deep Learning Models for Identifying Anatomical Landmarks in Chest CT

**DOI:** 10.3390/diagnostics12081844

**Published:** 2022-07-30

**Authors:** Parisa Kaviani, Bernardo C. Bizzo, Subba R. Digumarthy, Giridhar Dasegowda, Lina Karout, James Hillis, Nir Neumark, Mannudeep K. Kalra, Keith J. Dreyer

**Affiliations:** 1Massachusetts General Hospital, Harvard Medical School, Boston, MA 02114, USA; pkaviani@mgh.harvard.edu (P.K.); sdigumarthy@mgh.harvard.edu (S.R.D.); gdasegowda@mgh.harvard.edu (G.D.); lkarout@mgh.harvard.edu (L.K.); james.hillis@mgh.harvard.edu (J.H.); mkalra@mgh.harvard.edu (M.K.K.); kdreyer@partners.org (K.J.D.); 2Mass General Brigham Data Science Office, Boston, MA 02114, USA; nir.neumark@gmail.com

**Keywords:** deep learning model, chest CT examination, suboptimal chest CT imaging

## Abstract

(1) Background: Optimal anatomic coverage is important for radiation-dose optimization. We trained and tested (R2.2.4) two (R3-2) deep learning (DL) algorithms on a machine vision tool library platform (Cognex Vision Pro Deep Learning software) to recognize anatomic landmarks and classify chest CT as those with optimum, under-scanned, or over-scanned scan length. (2) Methods: To test our hypothesis, we performed a study with 428 consecutive chest CT examinations (mean age 70 ± 14 years; male:female 190:238) performed at one of the four hospitals. CT examinations from two hospitals were used to train the DL classification algorithms to identify lung apices and bases. The developed algorithms were then tested on the data from the remaining two hospitals. For each CT, we recorded the scan lengths above and below the lung apices and bases. Model performance was assessed with receiver operating characteristics (ROC) analysis. (3) Results: The two DL models for lung apex and bases had high sensitivity, specificity, accuracy, and areas under the curve (AUC) for identifying under-scanning (100%, 99%, 99%, and 0.999 (95% CI 0.996–1.000)) and over-scanning (99%, 99%, 99%, and 0.998 (95%CI 0.992–1.000)). (4) Conclusions: Our DL models can accurately identify markers for missing anatomic coverage and over-scanning in chest CTs.

## 1. Introduction

As a result of its expanding applications, there has been a tremendous increase in the use of computed tomography (CT) (R3-3) in modern medicine across the entire world. In the United States, CT imaging rates increased from 204 per 1000 individuals in 2000 to 428 per 1000 in 2016 [1]. Although CT contributes to less than 11% of all imaging tests, it accounts for up to 70% of the overall radiation doses in radiology departments [2]. Due to concerns about radiation-induced carcinogenesis stemming from burgeoning use and radiation doses, regulatory authorities and stakeholder organizations have recommended the need for protocol optimization radiation dose reduction (R1-2) and radiation dose monitoring [3].

Although CT vendors have introduced several innovative technologies to help reduce radiation dose from improved scanner efficiency and advanced image reconstruction techniques, careful designing of scan protocols and appropriate scanning procedures are critical for radiation-dose optimization [4]. Apart from the CT acquisition parameters, patient centering, positioning, and the use of appropriate scan length play an important role in dose optimization. Scanning regions beyond the anatomic regions of interest increases radiation dose beyond the intended regions and risks of detecting incidental lesions. Conversely, an incomplete evaluation of the entire volume of interest can lead to repeat scanning, with overlapping acquisition leading to a higher radiation dose. For our study, we label the former as over-scanning and the latter as under-scanning to differentiate them from over-beaming and over-ranging. Over-beaming refers to the excess radiation dose from the penumbra of the X-ray beam falling beyond the active detector area. Over-ranging is an extension of scan length in helical CT that contributes radiation dose at the beginning and end of the scan range but does not result in image data.

Despite specific information on anatomic landmarks for appropriate scan length, both over-scanning and under-scanning are frequent and lead to suboptimal evaluation and/or excess radiation dose [5]. To assess the frequency and implication of over-scanning and under-scanning, a manual evaluation of images is time-consuming, subjective, inefficient, and expensive. Although there are prior reports [2,6,7] on the artificial intelligence (AI)-driven determination of scan coverage, our work differs from prior publications since it uses a physician-trained tool for determining scan coverage. Such tools can help physicians train AI models without any programming knowledge or experience to target areas of quality improvement and safety that the software or medical companies might not deem as critical revenue or profit-generating solutions. The empowerment capabilities can motivate such physicians to improve local quality and safety aspects of patient services they offer. (R1-3, 3-1, 3-11)

We used a novel deep learning (DL)-based vision software platform (Cognex Corporation, Natick, MA, USA) to trainDL models for recognizing key anatomic landmarks in the chest to determine over-scanning and under-scanning at the lung apices and bases in chest CT examinations (R3-1). We used multi-site, multi-scanner, and multi-protocol image data to train and test the artificial intelligence (AI) (R3-3) models to ensure generalizability. The purpose of our study was to train and test (R2.2.4) deep DL algorithms to assess over- and under-scanning in chest CT examinations from two quaternary and two community hospitals.

## 2. Materials and Methods

### 2.1. Ethical Approval and Disclosures

The retrospective study was approved by the human research committee of our institutional review board at Massachusetts General Brigham (protocol number: 2020P003950, approval date: 23 December 2020). The need for informed consent was waived. All methods were performed in accordance with the relevant guidelines and regulations (e.g., Declaration of Helsinki). The study followed the Health Insurance Portability and Accountability Act’s guidelines (HIPAA). Mass General Brigham received funding through a collaboration with Cognex Corporation. MKK has received unrelated research grants from Coreline Corporation, Riverain Tech (Miamisburg, OH, USA) and Siemens Healthineers (Erlangen, Germany). Other co-authors have no financial disclosures.

### 2.2. Patients and Scanners

Using keywords of “chest CT with contrast and/or without contrast” on a commercial natural language understanding-based search engine (mPower, Nuance Inc, Burlington, NJ, USA), we queried a multi-institutional radiology report search database to identify the first consecutive 428 chest CT examinations from adult patients (age > 18 years) scanned between January and March 2021. All patients had a routine chest CT as part of their standard of care. The most common clinical indications for chest CT examinations were cancer staging and response assessment, hemoptysis, persistent cough, and unresolved pneumonia. Non-routine chest CT examinations, such as for lung nodule follow-up, lung cancer screening, tracheal protocol, and CT pulmonary angiography, were excluded. However, we did not exclude CT with metal or motion-related artifacts.

The first chest CT of each patient was selected from January–March 2021. Thus, the final sample size was 428 patients (mean age 70 ± 14 years; male:female 190:260). All patients were scanned in one of the four hospitals (Quaternary hospitals: Brigham and Women’s Hospital, Massachusetts General Hospital; Community hospital: Newton Wellesley Hospital, North Shore Medical Center).

All patients underwent contrast or non-contrast chest CT on one of the six multidetector-row CT scanners. The scanner-wise distribution of patients was GE Healthcare (64–256-section multidetector CT, GE Discovery 750HD or GE Revolution, *n* = 277 patients), Siemens Healthineers (64/96-section multidetector-row, single- and dual-source CT, Siemens Definition Edge, Flash or Force, *n* = 146 patients), Canon Medical Systems USA (320-section multidetector-row, Canon Aquilion ONE, *n* = 3 patients), and Philips Healthcare (64-multidetector-row single-source CT, *n* = 2 patients). All chest CT examinations were performed using routine chest CT protocols with helical scan mode, automatic tube voltage selection (on some scanners and 100–120 kV on scanners without the technique), automatic exposure control, 0.9:1 pitch, 0.4–0.5 s gantry rotation time, 1–1.25 mm reconstructed section thickness, 0.625–0.8 mm section overlap, 512 matrix, (R1-2.2) standard-to-medium reconstruction kernels, and with commercial vendor-specific iterative reconstruction techniques.

### 2.3. Data Partitioning

Upon importing the CT images, the deep learning platform automatically divides the data into training and testing datasets with either a 70–30 random distribution (R1-2-3) or a user-specified proportion (in an increment of 10%). In addition, for external validation, we also partitioned the data such that training data came from the two quaternary hospitals and CT data from the two community sites represented the testing dataset.

### 2.4. DL Model

The DL models were created using the Cognex Vision Pro Deep Learning software (Cognex Corporation). The software provides an intuitive user interface for training and testing models using a complete machine vision tool library with advanced DL techniques. The users or physicians without technical or programming experience can train DL models for specific clinical applications on the platform. The software offers four DL architectures that perform classification, segmentation, object location, and optical character recognition. The Green Tool helps classify objects based on training with labeled images from various classes, and the trained model can then be evaluated with new images that are not part of the training set. The tool can be used in high-definition mode (HDM) and focused mode. The focused mode used in our study does not display loss function, as there is no holdout validation set, so users can focus on the external test set performance as evidence that there is no overfitting of the model. (R2.2.4). It is comparable to classification neural networks such as DenseNet, ResNet, or Visual Geometry Group (VGG) but uses proprietary architectures and pre-processing steps to obtain improved performance in some settings. Further details of the architecture of the platform have been described in prior publications [8,9,10].

The segmentation tool is comparable to a semantic segmentation neural network (such as UNet) and is used for segmentation and detection of specific findings in images. Figure 1 shows the transverse chest CT images at the different section of the lung selected for model development.

At the time of writing this manuscript, the software is able to process a limited number of 2D images per patient as input, so we elected to train two different models on instances near the lung apices and bases (three images each at lung apices and bases) instead of including every instance in the CT series. At the apex, we de-identified and exported three images, including the cranial-most image (first image), an image just above the lung apices, and an image just below the lung apices (including a portion of lung apex/apices). Likewise, at the base from the same CT examinations, we exported three additional images, including the last, caudal-most image, an image just below the lung base, and an image including the lung base. Thus, for each CT, we exported and labeled six images from each CT examination included in our study (R3-5). Images just above the apex and just below the lung base represented the anatomic location of “optimal” scan coverage. Images below the apex and/or above the lung bases represented “incomplete” scan coverage. Finally, the first and last images represented “over-scanned” anatomy when these were >2.5 cm beyond the lung apices or bases. We recorded the distance between the first image and the image just above the lung apex (cranial distance) as well as the distance between the last image and the image just below the lung base (caudal distance).

All six CT images per exam were then uploaded to the DL building platform to train two DL models from scratch (R3-7,5) to recognize incomplete scan coverage from optimum scan coverage (DL-IA, incomplete coverage at lung apex; DL-IB, at lung base) and to differentiate over-scanned anatomy from optimum coverage (DL-OA, over-scan at lung apex; DL-OB, at lung base).

### 2.5. Training

We used the classification tool to classify three labeled CT images for each anatomical landmark at the lung apices and bases and trained two different DL models on the DL building platform, one (R3-5) each for lung apices and bases. Figure 2 shows the snippet of the software GUI used for classification. DICOM (Digital Imaging and Communications in Medicine) (R3-3) format CT images were first deidentified, uploaded into the software, and then labeled with each category: the first image, just above the apices, and the image with the apices for the apices detection model; the last image, just below the bases, and the image with the bases for the bases identification model). The training set in each model comprised of 70% of the image dataset selected randomly by the software, with the remaining 30% being the test dataset. Each model was cross-validated with a five-fold cross-validation method in which the test and training data were randomized, and the performance was reassessed.

We used the segmentation tool to segment the lung parenchyma using the “fill tool” functionality to isolate and train the model in identifying the lung regions with similar attenuation. Figure 3 shows a snapshot of the lung parenchyma mask used to train a segmentation model using Cognex software. A coinvestigator reviewed all images and labeled them as optimal (images without lung parenchyma) and suboptimal (images with lung parenchyma). The performance of each model was assessed using a five-fold cross-validation technique in which the test and training data were randomized.

### 2.6. Statistical Evaluation

The statistical analysis was performed using Microsoft Excel (Microsoft Inc., Redmond, WA, USA) and SPSS version 24 (IBM Inc., Chicago, IL, USA). To assess the DL performance, we compared the DL performance against the radiologist labeling of anatomic landmarks at the lung bases and apex (R3-8). We predefined true positive (radiologist and DL output agreed on image labels with the lung apices or bases included in the image), true negative (both radiologist and DL agreed on image labels, i.e., lung apices or bases not included in the image), false negative (disagreement with radiologist labeled apex or base not identified by DL), and false positive (DL misidentified lung base or apex on images labeled by the radiologist as without the lungs). The sensitivity, specificity, accuracy, and receiver operating characteristics (ROC) analysis with the area under the curve (AUC) were estimated for each DL model (at the lung bases and lung apices) with the five-fold cross-validations using Microsoft Excel and SPSS.

## 3. Results

The mean cranial (distance between the first image and the image just above the lung apices) and caudal distances (distance between the last image and image just below the lung bases) were 19.8 ± 7.9 mm and 40.5 ± 22.5 mm, respectively. There was a significant difference between the extent of over-scanning below the lung bases compared to over-scanning above the lung apices (*p* < 0.001).

The performance of both the classification and segmentation tools for differentiating incomplete coverage and over-scanning from optimum coverage at both the lung apices and bases images were identical in terms of sensitivity, specificity, accuracy, and AUC (*p* > 0.9). Recognition of anatomic landmark with the DL model at the lung apices and bases allowed us to automatically determine if there was adequate scan coverage or if there was under- or over-scanning of chest (R3-9). The sensitivity, specificity, accuracy, and AUCs of DL-IA and DL-OA for external validation datasets (sites A and B as training dataset and sites C and D as test dataset) were 97.77%, 98.20%, 97.99%, 0.997 (95% confidence interval (CI) 0.992–1.000), respectively. The corresponding values for DL-IB and DL-OB were 100%, 100%, 100%, and 1.000 (95%CI 1.000–1.000), respectively. The sensitivity, specificity, accuracy, and AUC of five-fold cross-validations of each DL model are summarized in Table 1, Table 2, Table 3 and Table 4. Figure 4 illustrates the AUC curve for the first cross-validation of each DL model (R3-10).

There was no significant difference in the performance of the DL models between male (AUC: DL-IA 0.988; DL-OA 0.978; DL-IB 1.000; DL-OB 0.1000) and female patients (AUC: DL-IA 0.987; DL-OA 0.979; DL-IB 1.000; DL-OB 1.000) (*p* > 0.1). Likewise, there was no significant difference in DL performance across the different CT scanner or section thickness (1–1.25 mm) types (*p* > 0.9). When classified into two age groups (≤40 years, >40 years), there was no significant difference in the performance of the DL-IA, DL-OA, DL-IB, and DL-OB between the two age groups (*p* > 0.1).

## 4. Discussion

The physician-trained DL models in our study accurately classified image levels suggestive of under- and over-scanning for routine chest CT protocols (98–100% accuracy). The performance of the DL models did not vary with patient age, gender, scanner types, section thickness, or external validation site data sources. The performance of the DL models for determining over-scanning is similar to other DL applications reported in prior publications. For example, Colevray et al. reported 0.99 accuracy for estimating over-scanning in chest CT examinations with a convolutional neural network (CNN)-based DL algorithm developed with 250 chest CT examinations [6]. The authors reported an over-scanning rate of 22.6% in 1000 chest CT examinations assessed with their algorithm. Another study from Salimi et al. reported that over-scanning particularly in the inferior direction occurred in more than 95% of chest CT examinations [7]. Another study from Demircioglu et al. reported a DICE score of 0.99 ± 0.1 for DL and radiologists’ annotations of scan range in routine chest CT [8]. Demircioglu et al. also reported a DICE score of 0.976 for such comparison between U-Net framework-based neural network and radiologists’ annotation of scan range for lung cancer screening protocol CT examinations [8].

As opposed to DL algorithms reported in the abovementioned studies [7], the model in our study detects under-scanning (missed scan coverage) in addition to over-scanning. To our best knowledge, no prior studies have reported on the use of DL for identifying missed anatomic coverage, which can lead to either incomplete diagnostic evaluation or trigger additional scanning over the missed anatomic region. The additional scanning often overlaps with the scan coverage of prior acquisition and therefore leads to higher radiation dose compared with a single scan acquisition without overlapping helical runs.

The primary implication of our study pertains to the ability of physicians to train DL models without any prior training or experience in deep learning, artificial intelligence, programming languages, and coding. Since radiation dose and service quality remain critical areas of concern for hospitals and imaging departments, we believe that DL models assessing proper image acquisition provides significant value to patients and healthcare systems. Some CT vendors now provide AI-enabled selection of scan start and end locations [6]. DL models such as the one used in this study can automatically monitor the relative performance of CT technologists and AI-based prescription of scan coverage. Such monitoring can lead to implementation of mitigating steps to correct frequent errors in prescribed scan length. In fact, Salimi et al. reported a 21% reduction in radiation dose for chest CT examinations with the use of a fully automated scan range definition from localizer or planning radiographs as compared to the manual prescription of scan length [7]. More work is, however, needed to develop and validate applications of such DL algorithms for identifying inappropriate scan coverage for other CT protocols, including those beyond the chest. At the same time, we believe that our DL models and work describes an important aspect of quality improvement in key imaging modality like CT, where physicians without prior deep learning experience can make use of DL building platform to address key problem areas that are either local or not of interest to the industrial-scale DL professionals. (R3-11)

Our study has limitations. First, we did not perform a power analysis to determine the required sample size although the observed performance of the AI model suggests that we had enough training and test cases. Second, we did not specifically enrich or down-sample chest CT examinations with and without abnormalities. It is possible, therefore, that a different type of datasets or abnormality spectrum will yield different results. However, we do not anticipate a substantial change in DL performance since we did not exclude chest CT examinations with apical or basilar pulmonary or pleural diseases. Third, we did not compare the DL model used in this study to other reported models since we did not have access to those models. Fourth, patient privacy rules forbid us to share our imaging datasets on an open-access platform. We have, however, uploaded the Excel datasheet of unidentifiable information on the GitHub website (https://github.com/parisak88/Chest-anatomical-landmark, accessed on 11 July 2022). Fifth, we did not assess the impact of artifacts, such as from metallic spinal prosthesis or patient motion, on the performance of the DL model. However, about 10–20% of our routine chest CTs have some motion artifacts, and therefore, our results should hold in presence of such artifacts. Sixth, we excluded patients who underwent CT for non-routine and non-chest protocols, and therefore, we cannot comment on the performance of our algorithm in patients scanned with the excluded protocols. Seventh, the DL model can classify chest CT for under-scanning and over-scanning but cannot determine the actual length in centimeter of over-scanned or missed anatomic lengths. Eighth, we did not use open-access CT datasets for our research since we could include data from four sites and multiple CT vendors and scanner types from our healthcare organization. Given the high performance of our DL model regardless of the scanners and vendors, we believe that there was no critical need for open access CT data in our study (R3-6). Finally, the performance of the DL model might have been artificially high on very limited set of images used for model training and testing. Thus, the results might not hold when a larger and more diverse set of images are exposed to the DL model.

## 5. Conclusions

In conclusion, the physician-trained deep learning-based models can identify missing anatomic coverage and over-scanning on images of routine chest CT examinations. Such determination can help audit the frequency and impact of under-scanning and over-scanning on diagnostic information and radiation doses associated with routine chest CT protocols. The DL-based identification of anatomic landmarks suggestive of under-scanning or over-scanning can help assess the quality and safety of CT scanning practices and can be leveraged to provide automated feedback to CT technologists so that errors in scan length can be reduced.

## Figures and Tables

**Figure 1 diagnostics-12-01844-f001:**
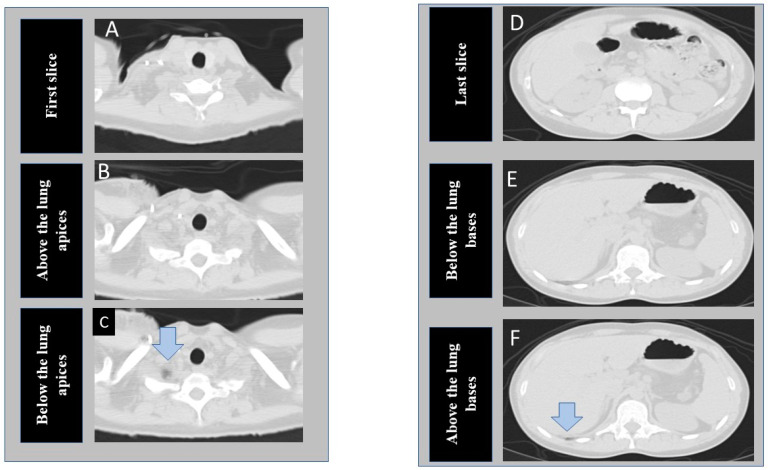
Transverse CT images at the superior (**A**–**C**) and inferior (**D**–**F**) scan locations of a chest CT examination. The first column shows the cranial-most (first image **A**), perfect scan (**B**), and under-scanned (**C**) image locations at the lung apex. The second column of images represent caudal most image (last image **D**), perfect scan location at lung base (**E**), and missed lung basilar anatomy (**F**). (The arrows point to tiny portions of lung parenchyma).

**Figure 2 diagnostics-12-01844-f002:**
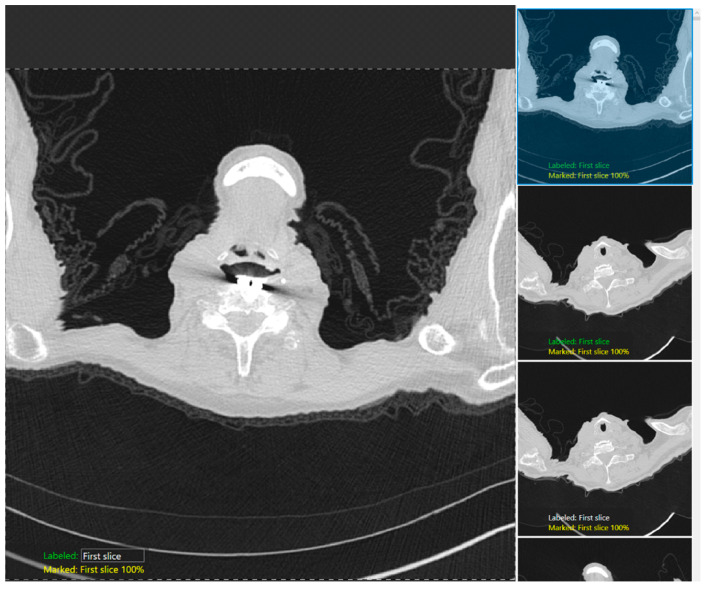
A snapshot of the Green Tool of the DL software used for classification of the anatomic locations at the lung apex.

**Figure 3 diagnostics-12-01844-f003:**
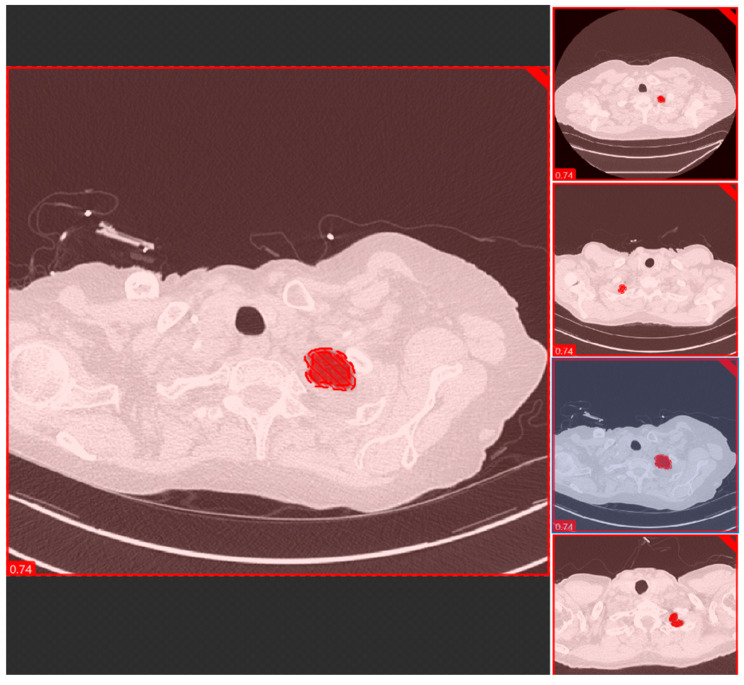
A snapshot of the Red Tool of the DL software used for segmenting and classification of the anatomic locations at the lung apex.

**Figure 4 diagnostics-12-01844-f004:**
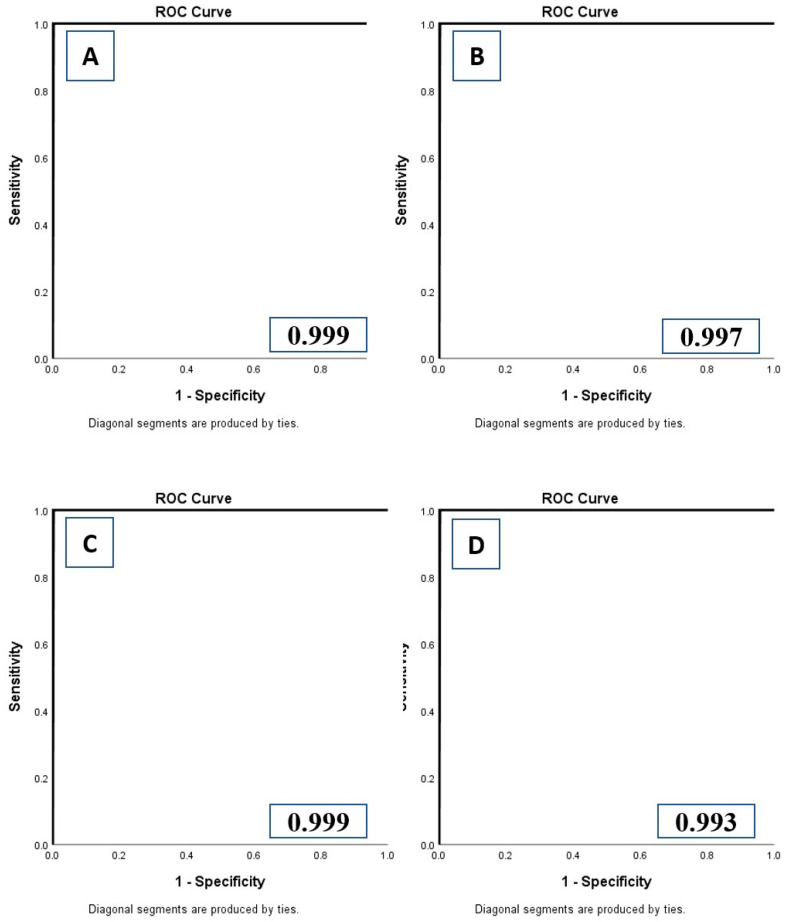
Receiver operating characteristic analyses with area under the curve (AUC) for DL-IA (**A**), DL-OA (**B**), DL-IB (**C**), and DL-OB (**D**) models.

**Table 1 diagnostics-12-01844-t001:** Results summary of five-fold, cross-validation of DL-IA model for detecting incomplete scan coverage at the lung apex (CI, confidence interval).

DL-IA	Sensitivity	Specificity	Accuracy	AUC	95% CI
First validation	100	99.77	99.85	0.999	0.996–1.000
Second validation	98.64	1000	99.55	0.998	0.993–1.000
Third validation	99.54	99.55	99.55	0.999	0.996–1.000
Forth validation	100	99.77	99.85	0.999	0.996–1.000
Fifth validation	100	99.11	99.40	0.996	0.991–1.000

**Table 2 diagnostics-12-01844-t002:** Results summary of five-fold, cross-validation of DL-OA model for detecting over-scanning at the lung apex (CI, confidence interval).

DL-OA	Sensitivity	Specificity	Accuracy	AUC	95% CI
First validation	100	98.66	99.33	0.997	0.993–1.000
Second validation	100	100	100	1.000	1.000–1.000
Third validation	100	98.25	99.11	0.999	0.996–1.000
Forth validation	98.22	100	99.11	0.996	0.991–1.000
Fifth validation	100	100	100	1.000	1.000–1.000

**Table 3 diagnostics-12-01844-t003:** Results summary of five-fold, cross-validation of DL-IB model for detecting incomplete scan coverage at the lung bases (CI, confidence interval).

DL-IB	Sensitivity	Specificity	Accuracy	AUC	95% CI
First validation	99.50	99.76	99.68	0.999	0.996–1.000
Second validation	99.01	99.76	99.52	0.998	0.992–1.000
Third validation	99.01	99.53	99.36	1.000	1.000–1.000
Forth validation	98.52	99.76	99.36	0.995	0.987–1.000
Fifth validation	100	100	100	1.000	1.000–1.000

**Table 4 diagnostics-12-01844-t004:** Results summary of five-fold, cross-validation of DL-OB model for detecting over-scanning at the lung bases (CI, confidence interval).

DL-OB	Sensitivity	Specificity	Accuracy	AUC	95% CI
First validation	98.58	100	99.28	0.993	0.984–1.000
Second validation	100	100	100	1.000	1.000–1.000
Third validation	99.06	100	99.52	0.995	0.988–1.000
Forth validation	100	99.06	99.53	0.998	0.992–1.000
Fifth validation	99.06	100	99.53	0.998	0.992–1.000

## Data Availability

The AI output presented in this study are openly available in [repository name chest-anatomical-landmark] at [https://github.com/parisak88/Chest-anatomical-landmark/blob/c4189ea2cee159c8ad5247f60d55b774229db9d0/de-identified%20EXCEL%20file%20containing%20information%20on%20model%20outputs%20for%20chest%20anatomical%20landmarks.xlsx] accessed on 11 July 2022, reference number [reference number].

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
