# Peer review of "Radiologist-Trained and -Tested (R2.2.4) Deep Learning Models for Identifying Anatomical Landmarks in Chest CT"

_diagnostics, 2022, doi:10.3390/diagnostics12081844_

Round 1

Reviewer 1 Report

Abstract

·       Please remove the information about the data.

·       Reorganize the paragraph with a better coherence among the sentences that have a much clear message about the motivation of the work.

Introduction:

·       Please mention why dose minimization is required, and the importance of it or its effect.

·       Please mention some related works, and highlight how this work differ significantly.

Materials and methods:

·       Section 2.2: Please mention clearly the temporal and spatial resolutions

·       Section 2.3:  70-30 data distribution is random or sequentially?

·       Section 2.4:

-        The title claims, deep learning model has been created, but in this section it is clearly mentioned, that a commercialized software is used. If a new model is created then please provide details of the algorithm, as it should be the main focus according to the title. Otherwise, if the same algorithm structure is used with different setting parameters than please change the title, it is not suitable as already an established algorithm is used.

-        Please provide a clear workflow chart, and mention the algorithms used and its architecture.

-        Please mention the final layer nodes, and which activation function is used.

Results:

·       Please provide a learning curve, to show there is no under sampling and oversampling issues. It cannot be solved by data distribution only. This is significant.

·       Use the mdpi formatting for the tables.

·       Correct the formatting in lines 271-273.

Conclusion:

Provide a separate section for the conclusion.

References:

·       Follow the mdpi guidelines for the formatting.  

·       Some of the references are more than 10 years old. Provide more recent ones.

Reviewer 2 Report

Note:

1. Figures 2, 3. Fuzzy text in the figure. Maybe make pointers with clear text.

Reviewer 3 Report

Radiologist-created deep learning models for identifying anatomic landmarks in chest CT

The authors have reported the application of Cognex VisionPro Deep Learning-based image analysis software to create four deep learning algorithms. The models were trained to recognize anatomic landmarks and classify chest computed tomography as optimum, under-scanned, or over-scanned scan lengths above and below lung apices and bases. Unfortunately, I have manifold challenges with the manuscript as follows.

1. The main contribution of the study is not apparent in the text and the specific problem being addressed is unclear.

2. The authors claimed the creation of four deep learning models in the second statement in the abstract. However, in the second to the last sentence of the abstract, they mentioned that the two deep learning models had high sensitivity, specificity, and so on which is confusing.

3. Certain acronyms such as CT, DL, CI, AI, GE, GUI, DICOM, DL-IA, DL-IB, DL-OA, and DL-OB were introduced in the manuscript without being defined. 

4. The literature review is incomprehensible with a lack of sufficient previous research studies on the topic. Many relevant articles from Web of Science, Scopus, and Nature databases were not cited to reinforce the claims in the manuscript. 

5. The manuscript has suffered from methodological flaws that have made discussions ill-founded. The deep learning methods were not comprehensively described, and it is unclear how the radiologists created the deep learning models. The model architectures and parameters were not defined which makes it hard to understand how the models work.

6. The experimental datasets on the computed tomography examinations were not described anywhere in the manuscript and it is difficult to access the datasets. There are open-access chest computed tomography datasets that could have been used to complement the private datasets for validating the deep learning models.

7. It is unclear whether the deep learning models were created from the scratch or they are components of a suite of libraries in Cognex.

8. It is not clear how the statistical evaluation was done using Microsoft Excel and SPSS and a description of experimentation was not provided.

9. The results of the study are of little interest. There is a lack of a contextual analysis of the experimental data explaining the performances of deep learning models.

10. It was mentioned in section 2.6 that the sensitivity, specificity, accuracy, ROC, and AUC were estimated for each deep learning model. However, the ROC graphs are not presented, and the results are not compared with those of the existing deep learning algorithms used in the existing related studies.

11. The discussion did not connect to the literature to provide an argument that fills the existing gap and addresses the problem under investigation.

Round 2

Reviewer 3 Report

This version of the manuscript is far better than the previous one.